# Characterizing the transport and utilization of the neurotransmitter GABA in the bacterial pathogen *Brucella abortus*

**James A. Budnick**[1,2], **Lauren M. Sheehan**[1,2], **Angela H. Benton**[1,2], **Joshua E. Pitzer**[3], **Lin Kang**[2,4], **Pawel Michalak**[2,4,5], **R. Martin Roop, II**[3], **Clayton C. Caswell**[1,2]*

**1** Department of Biomedical Sciences and Pathobiology, Virginia-Maryland College of Veterinary Medicine, Blacksburg, Virginia, United States of America, **2** Center for One Health Research, Blacksburg, Virginia, United States of America, **3** Department of Microbiology and Immunology, East Carolina University Brody School of Medicine, Greenville, North Carolina, United States of America, **4** Edward Via College of Osteopathic Medicine, Blacksburg, Virginia, United States of America, **5** Institute of Evolution, University of Haifa, Haifa, Israel

* caswellc@vt.edu

**Data Availability Statement:** All RNA-seq files are available from the NCBI Sequence Read Archive (SRA) database (accession number PRJNA629010)

## Abstract

The neurotransmitter gamma-aminobutyric acid (GABA) is the most abundant inhibitory neurotransmitter in the human brain; however, it is becoming more evident that this non-proteinogenic amino acid plays multiple physiological roles in biology. In the present study, the transport and function of GABA is studied in the highly infectious intracellular bacterium *Brucella abortus*. The data show that $^3$H-GABA is imported by *B. abortus* under nutrient limiting conditions and that the small RNAs AbcR1 and AbcR2 negatively regulate this transport. A specific transport system, *gts*, is responsible for the transport of GABA as determined by measuring $^3$H-GABA transport in isogenic deletion strains of known AbcR1/2 regulatory targets; however, this locus is unnecessary for *Brucella* infection in BALB/c mice. Similar assays revealed that $^3$H-GABA transport is uninhibited by the 20 standard proteinogenic amino acids, representing preference for the transport of $^3$H-GABA. Metabolic studies did not show any potential metabolic utilization of GABA by *B. abortus* as a carbon or nitrogen source, and RNA sequencing analysis revealed limited transcriptional differences between *B. abortus* 2308 with or without exposure to GABA. While this study provides evidence for GABA transport by *B. abortus*, questions remain as to why and when this transport is utilized during *Brucella* pathogenesis.

## Introduction

Gamma-aminobutyric acid (GABA) is a non-proteinogenic amino acid that is a common and important inhibitory neurotransmitter in the vertebrate brain [1]. However, our understanding of the biological function of GABA has expanded over the years to include neurobiology, immunology, and bacteriology. With regards to metabolism, the GABA shunt is utilized by both prokaryotes and eukaryotes to metabolize GABA to succinate, which can then be

**Funding:** C.C.C. AI125958 National Institute of Allergy and Infectious Diseases (NIAID) https://www.niaid.nih.gov/ The funders had no role in study design, data collection and analysis, decision to publish, or preparation of the manuscript.

**Competing interests:** The authors have declared that no competing interests exist.

supplied into the TCA cycle [2, 3]. This is achieved by transport of exogenous GABA or conversion of endogenous glutamate to GABA by the enzyme glutamate decarboxylase (GAD). GAD is an important enzyme for both the production of GABA and deacidification of the intracellular environment. If the pH of the cell becomes unfavorably low, GAD can convert glutamate to GABA with the attachment of a proton, then export it out of the cell, which will result in increased intracellular pH [4, 5].

In plants, several studies have revealed the necessity for GABA during metabolism and developmental growth [6–8], but GABA is also an important modulator of immunity against pathogenic organisms, including insects, fungi, and bacteria. Upon plant cell damage, the pH of the plant intracellular environment will decrease, activating the GAD system, producing an excess of secreted GABA surrounding the damaged area of the plant [3]. In insects, increased environmental GABA concentrations have been shown to lead to decreased larvae growth rate, survival, and feeding by pests on tobacco plants [9–11]. Exogenous GABA also has a negative effect on bacterial pathogenesis of plants. A deletion of the GABA transaminase, responsible for the conversion of GABA to succinic semialdehyde in the GABA shunt, in *Pseudomonas syringae* led to decreased expression of a type III secretion system required for full virulence of the bacterium [12]. This deletion strain displayed significant reductions in virulence *in planta* when compared to the parental strain, which was attributed to the decreased expression of the type III secretion system [12]. Decreased virulence by exogenous GABA has also been shown in the bacterial plant pathogen *Agrobacterium tumefaciens*. *A. tumefaciens* encodes two ABC transport systems, Bra and Gts, that import exogenous GABA [13, 14]. Once GABA is transported into *A. tumefaciens*, it is catabolized via the GABA shunt and byproducts of the shunt induce the expression of AttM, a lactonase [15]. This lactonase will quench quorum signaling molecules expressed by *A. tumefaciens* leading to a decrease in the expression of virulence related genes [13, 14]. By increasing the expression of GAD and secretion of GABA by plants, studies have shown that tobacco plant susceptibility to *A. tumefaciens* can be decreased. Mutating the GAD system in a plant, however, led to increased T-DNA transfer, a major virulence factor, from *A. tumefaciens* to a tomato plant model [15, 16]. Alternatively, increasing GABA transaminase activity in *A. tumefaciens*, causing a decrease in intracellular GABA concentrations, led to higher rates of T-DNA transfer and transformation of tomato plants, further emphasizing the inhibitory role of GABA in *A. tumefaciens* virulence [16].

More recently, GABA has been observed to be an immunomodulator in mammalian systems, and several studies have shown that GABA activates immune cells and plays a role in the antimicrobial activity of macrophages. Bhat et al. demonstrated that immune cells (dendritic cells and macrophages) can synthesize and catabolize GABA, and the presence of GABAergic agents led to a decrease in inflammation [17]. The authors hypothesize that GABA could potentially be utilized as a signaling molecule between immune cells to modulate inflammation. Interestingly, GABAergic signaling has also been shown to enhance phagosomal maturation in macrophages, and inhibition of this signaling led to increased intracellular concentrations of bacteria within a macrophage [18]. These studies reinforce that further understanding of the role GABA plays in eukaryotic immunology is necessary.

*Brucella* spp. are pathogenic intracellular bacteria within the Order *Rhizobiales* in the Class *Alphaproteobacteria*. The brucellae infect a variety of mammalian species, both wild and domesticated, in which brucellosis primarily affects reproductive health in these animals, and chronic infection can lead to multiple organ complications [19, 20]. Several *Brucella* spp. also have the capacity to cause infection in humans via direct contact with contaminated animal products, and brucellosis is one of the most prevalent zoonoses worldwide [21]. Human brucellosis primarily presents as flu-like symptoms including an undulating fever and chronic infection can also cause damage to multiple organs [22, 23]. *Brucella* spp. are stealth pathogens

that contain few classical virulence factors and primarily evade the host immune system by adaptation to the harsh intracellular environment and formation of a replicative niche within primary immune cells (dendritic cells and macrophages) of the host [24, 25].

Small regulatory RNAs (sRNAs) are understudied virulence factors of the brucellae, and importantly, sRNAs can swiftly regulate gene function post-transcriptionally to adapt to changing environmental conditions [26]. While characterizing the role of the sRNAs AbcR1 and AbcR2 (AbcR1/2) in *B. abortus* pathogenesis, it was demonstrated that these small RNAs primarily function as negative regulators of several ABC type transport systems in *B. abortus* [27]. Two loci regulated by AbcR1/2 have previously been studied with regards to GABA transport in other *Alphaproteobacteria* [28, 29]. One putative transport system, a locus including *bab1_1792-bab1_1799* (*bab_rs24455-bab_rs24485)*, encodes proteins with high amino acid sequence identity to one of the GABA ABC transport systems, Bra, mentioned above in *Agrobacterium tumefaciens* (Fig 1A). It should be noted that the *B. abortus* genome has recently been reannotated, and while the old nomenclature will be utilized throughout this manuscript, new gene designations (*bab_rs#####*) will follow the old designation after the initial mention in the manuscript for reference. Similar to *B. abortus*, the homologous transport system in *A. tumefaciens* has also been shown to be negatively regulated by AbcR1 [28]. The second putative transport system, a locus including *bab2_0876-bab2_0879* (*bab_rs30470-bab_rs30485)*, encodes proteins with low amino acid sequence identity to the GABA Transport System, Gts, in *Rhizobium leguminosarum* and *A. tumefaciens* (Fig 1B) [29].

No previous studies have explored the function of GABA within the brucellae, with the exception of the GAD system [30]. Interestingly, the functionality of the GAD system differs between species of *Brucella*. The "classical" species of *Brucella* (*B. melitensis*, *B. abortus*, *B. suis*, *B. canis*, *B. neotomae*, and *B. ovis*) do not possess a functional GAD system due to point or frame-shift mutations in *gadB* and *gadC* genes [30]. Thus, the potential role for GABA utilization by the "classical" species of *Brucella* is unknown. The following study will focus on characterizing the potential import of GABA into *B. abortus* and elucidate the functional role of GABA in *Brucella* pathogenesis. The results reveal that GABA is transported under nutrient limiting conditions, and GABA transport is regulated by AbcR1 and AbcR2 in *B. abortus*. The data also showed minimal metabolic or regulatory potential for GABA by *B. abortus in vitro* under the conditions tested.

## Results

### *B. abortus* can import $^3$H-GABA, and this transport is inhibited by the presence of glutamate

GMM is a commonly used minimal medium in which to grow *Brucella* to mimic a nutrient-limiting environment [31]. *Brucella* growth is sustained in this medium but will not reach high concentrations compared to growth in nutrient rich medium, such as brucella broth. GMM specifically contains the amino acid glutamate as a carbon and nitrogen source. We hypothesized that if *Brucella* could transport GABA, it would most likely occur in growth of limited nutrient concentrations when transport system expression is increased. A GABA transport study was utilized to determine 1) if *B. abortus* could import $^3$H-GABA in GMM and 2) if glutamate in the medium would inhibit the uptake of $^3$H-GABA. The experiment was conducted with GMM containing glutamate (GMM) and GMM without the addition of glutamate (GMM-Glu).

Briefly, *B. abortus* strains were grown on SBA plates for 48 hours, and then cultures of *B. abortus* were incubated for 20 minutes in either GMM or GMM-Glu. Subsequently, the cultures were inoculated with $^3$H-GABA and incubated for an additional 20 minutes. Cultures

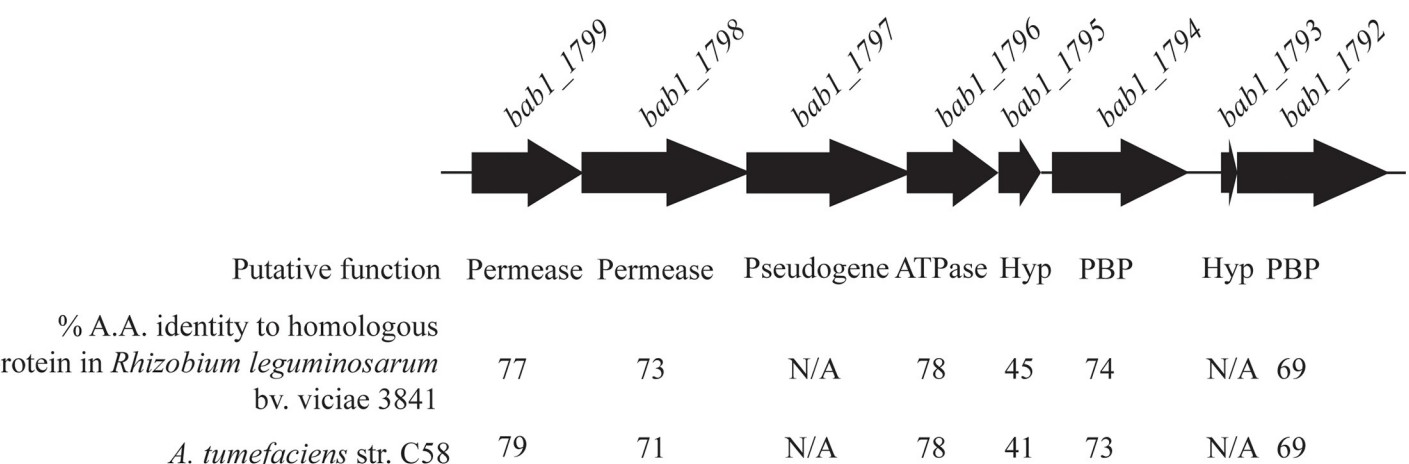

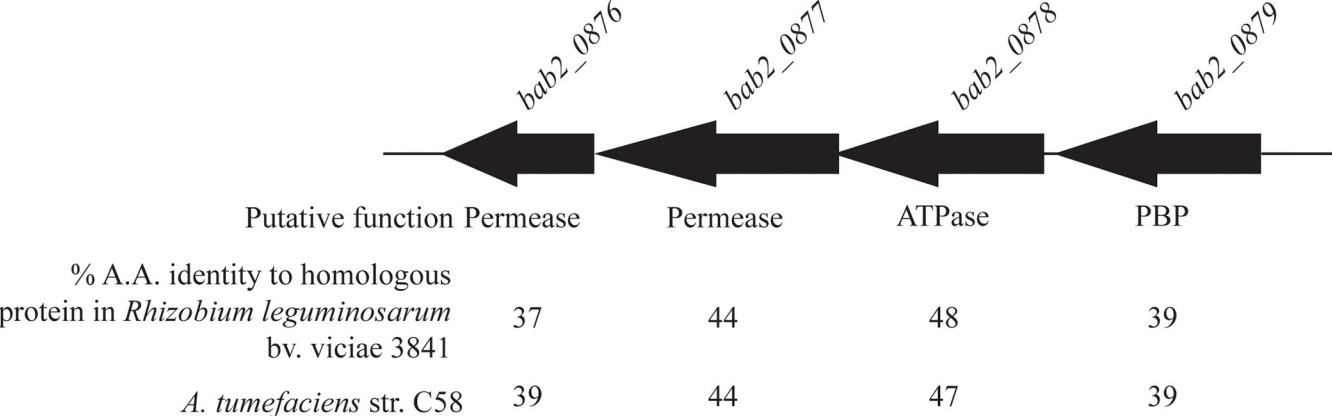

**Fig 1. Organization of putative GABA ABC-type transport systems in *B. abortus* 2308 and homology to the GABA transport systems in *Rhizobium leguminosarum* bv. viciae 3841 and *Agrobacterium tumefaciens* str. C58.** A. Genetic organization of *bab1_1972-bab1_1799* located on chromosome I of *Brucella melitensis* biovar Abortus 2308. Putative functions for each gene and percent amino acid identity to *bra* genes in related organisms are located below the gene. Proteins encoded from this locus exhibit high amino acid identity to the *bra* locus in *Agrobacterium tumefaciens* str. C58. B. Genetic organization of *bab2_0876-bab2_0879* located on chromosome II of *Brucella melitensis* biovar Abortus 2308. Putative functions for each gene and percent amino acid identity to *gts* genes in related organisms are located below the gene. Proteins encoded from this locus exhibit moderate amino acid identity to the *gts* locus in *R. leguminosarum* bv. viciae 3841.

were then collected via filtration through a syringe filter (see methods). The radioactivity of the filter was measured to quantify the amount of radiation imported by the brucellae collected. If $^3$H-GABA is imported by *B. abortus*, then the filter will measure high radioactivity above background; however, if $^3$H-GABA is not imported by *B. abortus*, then the $^3$H-GABA will pass through the filter and the filter will not measure high radioactivity above background.

As a control, 1000-fold excess non-radiolabeled GABA was added to the cultures simultaneously to out-compete $^3$H-GABA import and show specificity for GABA.

The assay revealed that *B. abortus* 2308 imported $^3$H-GABA in both GMM and GMM-Glu and this transport was specific for GABA as the addition of excess non-radiolabeled GABA in both culture media significantly decreased 3H-GABA import (Fig 2). However, when glutamate was present in the culture medium, the amount of $^3$H-GABA imported by *B. abortus* 2308 decreased by over 95%. These data indicate that transport of $^3$H-GABA is increased under nutrient limitation.

## The presence of proteinogenic amino acids does not considerably inhibit the import of $^3$H-GABA into *B. abortus*

The above $^3$H-GABA transport assay was again utilized to assess whether proteinogenic amino acids could competitively inhibit $^3$H-GABA import in *B. abortus*. The same assay was utilized, except that *B. abortus* 2308 was incubated in GMM-Glu (as glutamate inhibited $^3$H-GABA transport in Fig 2) throughout the experiment. After the initial incubation period, the cultures were inoculated with $^3$H-GABA and the addition of no inhibitor or 100 µM individual amino acids, resulting in a ratio of 1:1,000 $^3$H-GABA:nonradiolabeled amino acid. The cultures were filtered and measured for radioactivity. The control group showed $^3$H-GABA uptake was almost completely inhibited by the presence of 1,000-fold excess nonradiolabeled GABA (Fig 3). The import of $^3$H-GABA was not significantly changed by the presence of most other

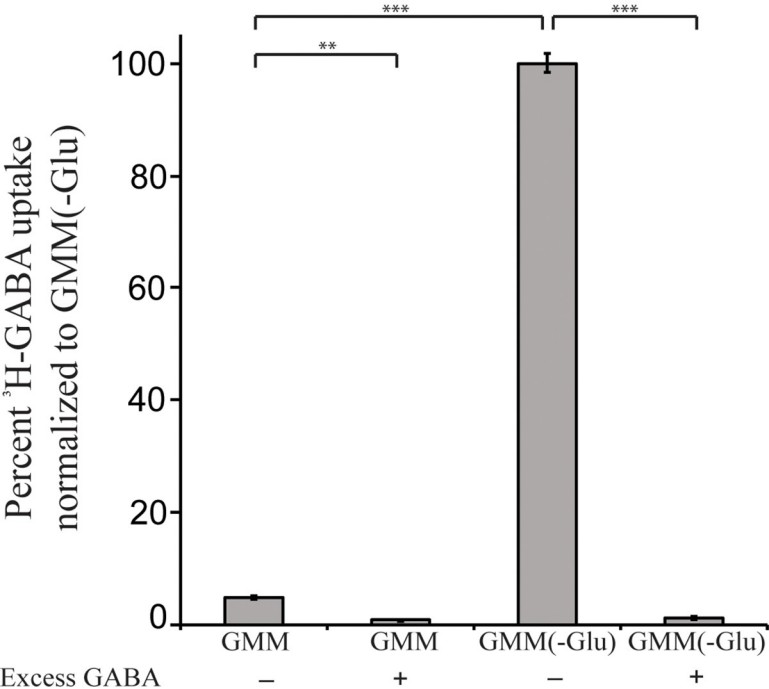

**Fig 2. $^3$H-GABA import is induced under nutrient limiting conditions.** $^3$H-GABA uptake by *B. abortus* 2308 was assessed in minimal medium with (GMM) and without (-Glu) the addition of glutamate to the medium. Data is normalized to GMM(-Glu) at 100%. Controls include the addition of excess nonradiolabled GABA to competitively inhibit $^3$H-GABA uptake. The asterisks denote a statistically significant difference (** $P < 0.005$, *** $P < 0.0005$; Student's t test) in uptake.

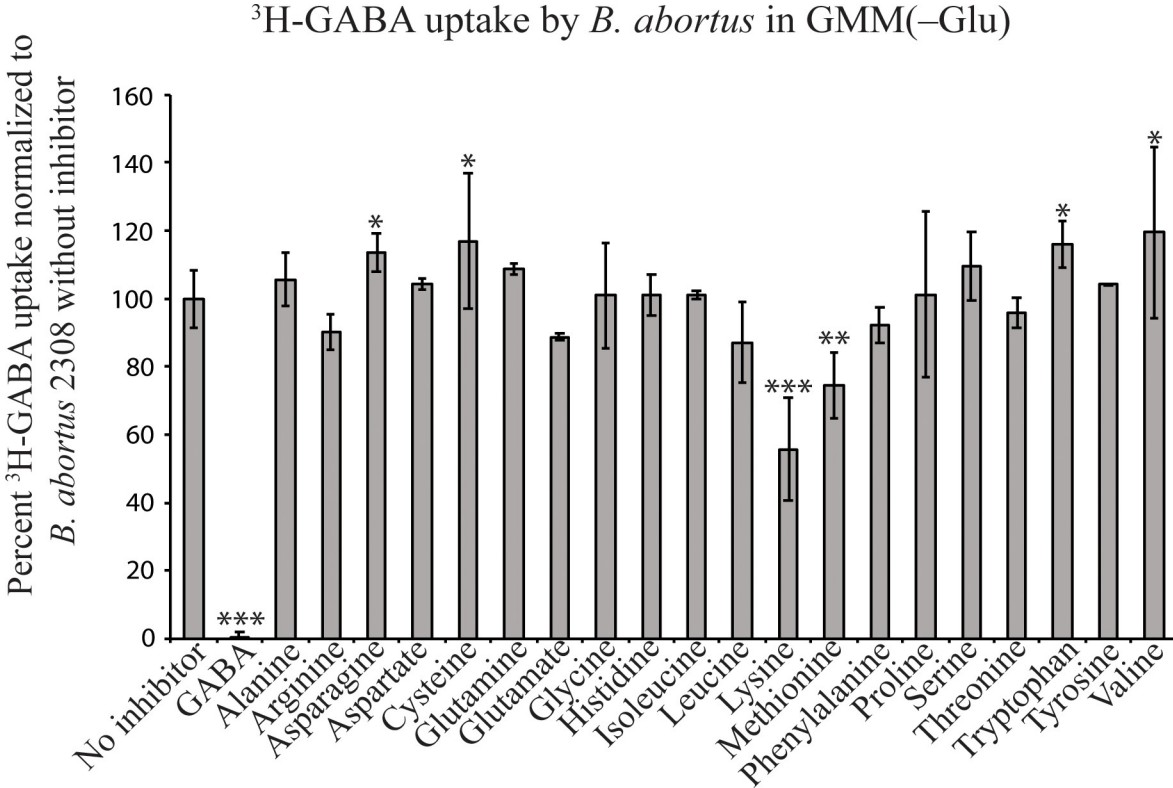

**Fig 3. ³H-GABA import by *B. abortus* 2308 is not greatly inhibited by the presence of other amino acids *in vitro*.** ³H-GABA uptake by *B. abortus* 2308 was assessed uninhibited and in the presence of 1,000-fold excess GABA or 20 proteinogenic amino acids. Data is normalized to the absence of inhibitor at 100%. The asterisk denotes a statistically significant difference (* $P<0.05$, ** $P<0.005$, *** $P<0.0005$; Student's t test) in uptake of ³H-GABA between *B. abortus* 2308 uninhibited (no inhibitor) and in the presence of excess nonradiolabled GABA, asparagine, cysteine, lysine, methionine, and tryptophan.

nonradiolabled amino acids. However, 1,000-fold excess lysine or methionine significantly decreased ³H-GABA import and 1,000-fold excess asparagine, cysteine, tryptophan, or valine significantly increased ³H-GABA import. It should be noted, however, that these difference are small, implying a preference for ³H-GABA import over all proteinogenic amino acids under these conditions.

Combined with the previous experiment, these results suggest that glutamate does not competitively inhibit the transport of GABA via interactions with the putative transport system, but rather the expression of the GABA transporter may be induced in the absence of glutamate. This experiment also indicates that the mechanism responsible for the transport of GABA would preferentially transport GABA prior to the transport of other amino acids; if this mechanism can transport other amino acids at all.

### ³H-GABA import is inhibited by the sRNAs AbcR1 and AbcR2

The regulation of GABA transport is mediated by the sRNA AbcR1 in *A. tumefaciens*, and a deletion of *abcR1* results in increased import of radiolabeled GABA in *A. tumefaciens* [28]. *B. abortus* 2308 encodes the sRNAs AbcR1 and AbcR2, homologs of AbcR1 in *A. tumefaciens*, which regulate ABC-type transport systems, including the homologous GABA transport in *A. tumefaciens* [27]. Therefore, we hypothesized that a deletion of *abcR1* and *abcR2* in *B. abortus* would result in increased GABA transport.

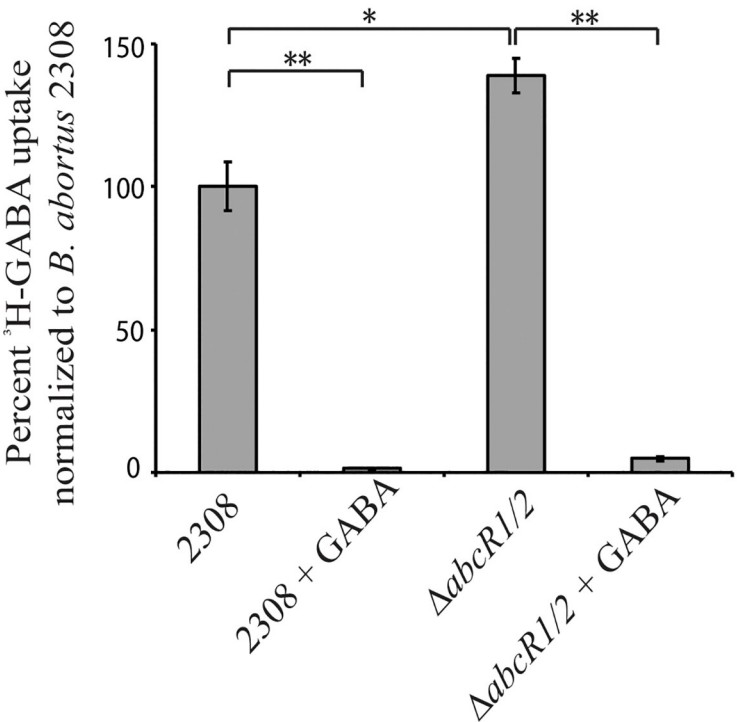

Fig 4. **$^3$H-GABA import is negatively regulated by the sRNAs AbcR1 and AbcR2 in *B. abortus*.** $^3$H-GABA uptake by *B. abortus* 2308 and *B. abortus* 2308:: Δ*abcR1/2* was assessed in minimal medium, GMM(-Glu). Data is normalized to 2308 at 100%. Asterisks denote a statistically significant difference (* P<0.05, ** P<0.005; Student's t test) in uptake.

To test this hypothesis, the above mentioned $^3$H-GABA transport assay was utilized to assess the import of $^3$H-GABA by *B. abortus* 2308 (2308) or *B. abortus* 2308::Δ*abcR1*Δ*abcR2* (Δ*abcR1/2*) (Fig 4). The results indicated that $^3$H-GABA import was increased by almost 50% in Δ*abcR1/2* compared to the parental strain. 1,000-fold excess nonradiolabled GABA was added to cultures as a control, which inhibited $^3$H-GABA import in both 2308 and Δ*abcR1/2*. This indicated that GABA transport was negatively regulated by the sRNAs AbcR1 and AbcR2, similarly to what has been observed in *A. tumefaciens*.

### *bab2_0879* is necessary for the transport of $^3$H-GABA in *B. abortus*

Due to the negative regulation of $^3$H-GABA transport by AbcR1/2 (Fig 4), it was hypothesized that one or more of the transport systems negatively regulated by AbcR1/2 is responsible for the transport of $^3$H-GABA. To test this hypothesis, the above $^3$H-GABA transport assay was repeated to measure $^3$H-GABA uptake in *B. abortus* 2308, as well as several strains carrying isogenic deletion of genes encoding putative periplasmic binding proteins of transporter systems that are significantly negatively regulated by AbcR1 and AbcR2 [27, 32]. These deletion strains include Δ*bab2_0612* (*bab_rs29240*), Δ*bab2_0879*, and Δ*bab1_1792-bab1_1794* (Fig 5). The isogenic deletion strains of *bab2_0612* and *bab2_0879* were constructed previously [32], and the combined *bab1_1792–1794* deletion strain was generated in the present study. The transport of $^3$H-GABA only decreased by ~20% in Δ*bab2_0612* and Δ*bab1_1792-bab1_1794*

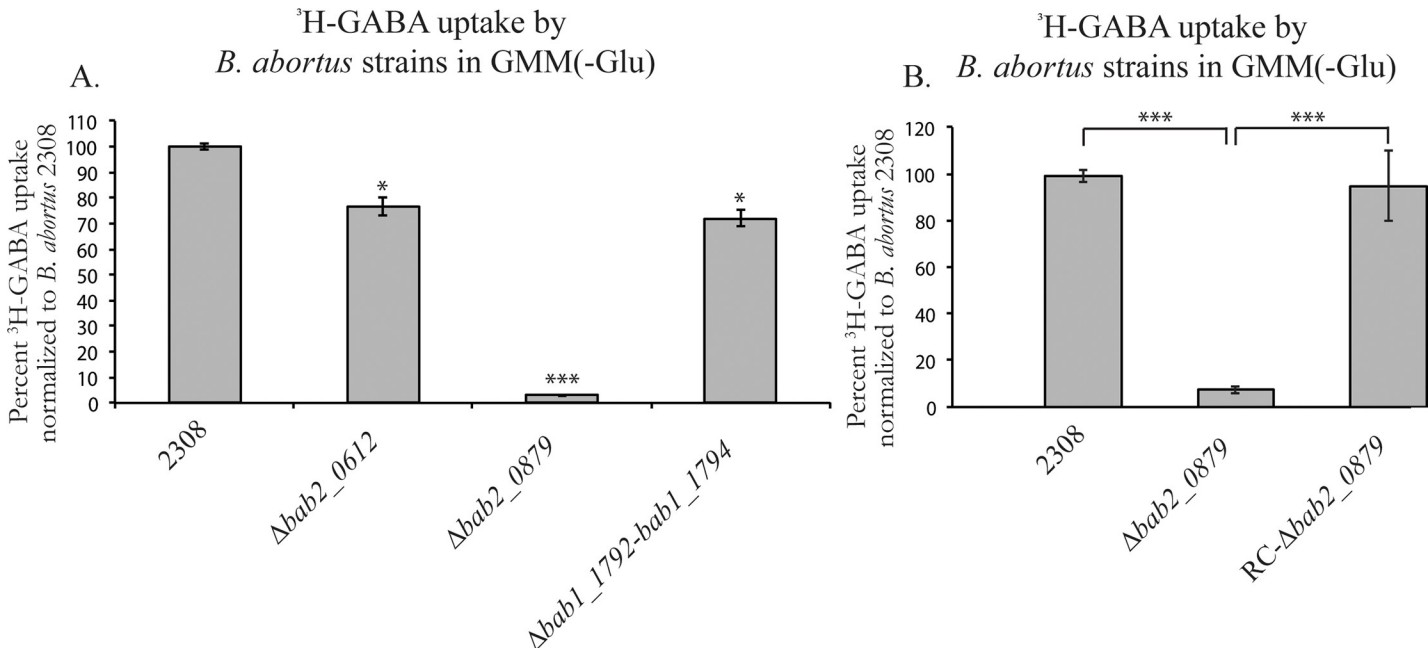

**Fig 5. Transport of ³H-GABA transport in *B. abortus* 2308 by AbcR regulated systems.** A. ³H-GABA uptake by *B. abortus* 2308, Δ*bab2_0612*, Δ*bab2_0879*, Δ*bab1_1792*Δ*bab1_1794*, and Δ*bab2_0879* in assessed in minimal medium, GMM(-Glu). Data is normalized to 2308 at 100%. Asterisks denote a statistically significant difference (* P<0.05, *** P<0.0005; Student's t test) in uptake between *B. abortus* 2308 and deletion strains. B. ³H-GABA uptake by *B. abortus* 2308, Δ*bab2_0879*, and Δ*bab2_0879*-RC*bab2_0879* in minimal medium, GMM(-Glu). Data is normalized to 2308 at 100%. Asterisks denote a statistically significant difference (*** P<0.0005; Student's t test) in uptake between strains.

when compared to the parental strain, *B. abortus* 2308, indicating that they may partially be involved in GABA transport. However, the transport of ³H-GABA by Δ*bab2_0879* decreased by ~97% when compared to *B. abortus* 2308, implicating *bab2_0879* as a component of the main transporter of GABA in *B. abortus* (Fig 5A). Reconstruction of *bab2_0879* on the *B. abortus* 2308:: Δ*bab2_0879* genome complemented ³H-GABA import (Fig 5B). These data clearly show that *bab2_0879* is involved in GABA transport and should be annotated as *gtsA* based on homology and function.

### *bab2_0879* is not necessary for survival and replication in peritoneal derived macrophages nor chronic infection of a mouse model of brucellosis via the oral route of infection

It was reported previously that a deletion of *bab2_0879* did not affect the ability of *B. abortus* to colonize the spleen of a mouse infected intraperitoneally [32]. Therefore, this strain was further tested for its ability to survive and replicate within peritoneally derived macrophages *in vitro* and to colonize the spleens of mice infected orally *in vivo*.

A gentamycin protection assay was utilized to assess the survival and replication of *B. abortus* strains within macrophages. Naïve macrophages were isolated from the peritoneal cavity and infected with either *B. abortus* 2308 or *B. abortus* 2308:: Δ*bab2_0879* (Δ*bab2_0879*) at an MOI of 100. Infected macrophages were lysed 2, 24, and 48 hours post-infection and serial diluted to calculate CFU brucellae/well. A deletion of *bab2_0879* did not affect the ability of *B. abortus* to survive and replicate within macrophages when compared to the parental strain *B. abortus* 2308 (Fig 6A).

BALB/c mice were infected orally with 10⁹ CFU of *B. abortus* 2308 or Δ*bab2_0879* and infection was monitored 1, 2, and 4 weeks post-infection. After 1, 2, or 4 weeks, the mice were

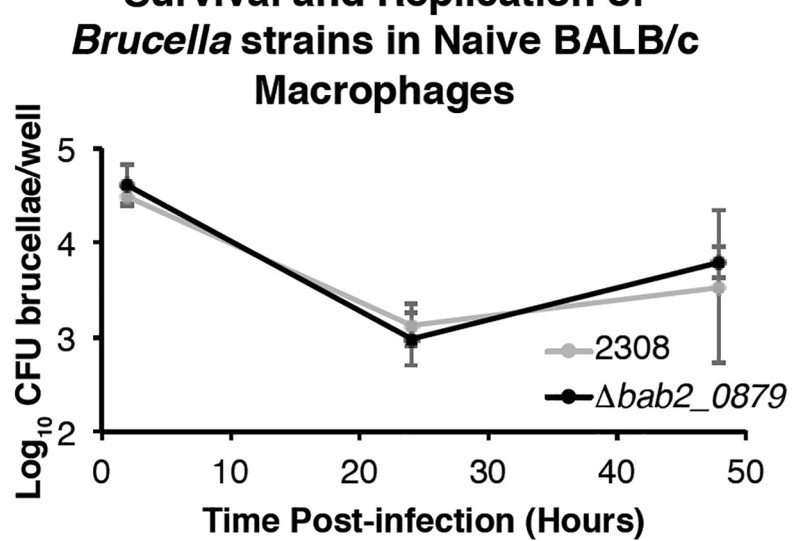

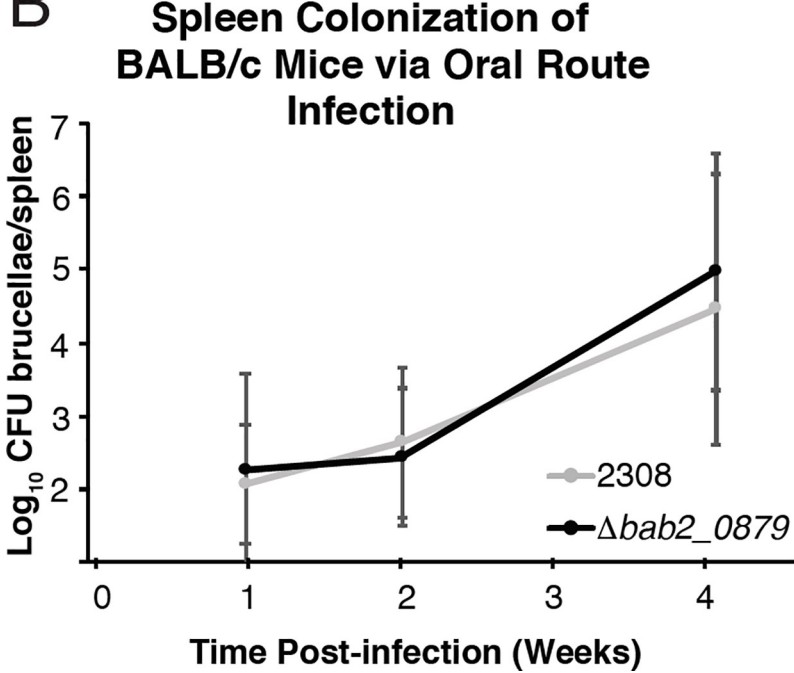

**Fig 6. Virulence of *B. abortus* 2308 and Δ*bab2_0879* in peritoneally derived macrophages and BALB/c mice.** A. Macrophage survival and replication experiments. Cultured peritoneal macrophages from BALB/c mice were infected with *B. abortus* 2308 and the isogenic *bab2_0879* deletion strain (Δ*bab2_0879*). At the indicated times post-infection, macrophages were lysed, and the number of intracellular brucellae present in these phagocytes was determined by serial dilution and plating on agar medium. B. Oral mouse infection experiments. BALB/c mice (6 per strain) were infected intraperitoneally with *B. abortus* 2308 and Δ*bab2_0879*. Mice were sacrificed 1, 2, and 4 weeks post-infection, and $Log_{10}$ brucellae/spleen were calculated. The data are presented as average numbers of brucellae ± standard deviations of results from the 6 mice (3 male and 3 female mice) colonized with a specific *Brucella* strain at each time point.

sacrificed, spleens removed and homogenized, and homogenates were serial diluted to determine CFU brucellae/spleen (Fig 6B). Under the conditions tested, the ability of the *B. abortus* Δ*bab2_0879* strain to colonize the spleen was not significantly changed when compared to the parental strain.

## GABA is not utilized as a nitrogen or carbon source by *Brucella*

To elucidate the biological role of GABA in the brucellae, two situations were considered: GABA is either acting as a source of carbon and/or nitrogen, or GABA functions as a signaling molecule to induce changes in gene expression. The hypothesis that GABA is a metabolite was first examined. As mentioned before, GMM is often utilized as a defined medium to mimic a nutrient-limiting environment. This medium was developed in 1958 by Philipp Gerhardt and contains several sources of carbon; including lactic acid, glycerol, and glutamate; and glutamate as the sole nitrogen source [31]. Growth curves were utilized to test the ability for GABA to be utilized as a nitrogen source for the brucellae via replacement of glutamate. *B. abortus* 2308 was grown overnight in brucella broth to late exponential phase, pelleted and washed, and then used to inoculate GMM with glutamate (GMM), GMM without glutamate (GMM-Glu), or GMM without glutamate but supplemented with GABA (0.15%) (GMM-Glu +GABA). Growth of the bacterium was measured in each culture for 175 hours (Fig 7A). Initially, all cultures showed growth, most likely due to residual nutrients from the nutrient rich brucella broth. However, *B. abortus* 2308 grown in GMM-Glu and GMM-Glu+GABA revealed a decrease in bacterial concentration in comparison to *B. abortus* 2308 in GMM over time. This indicated that GABA is likely not utilized as a nitrogen source in place of glutamate for sustained *B. abortus* growth.

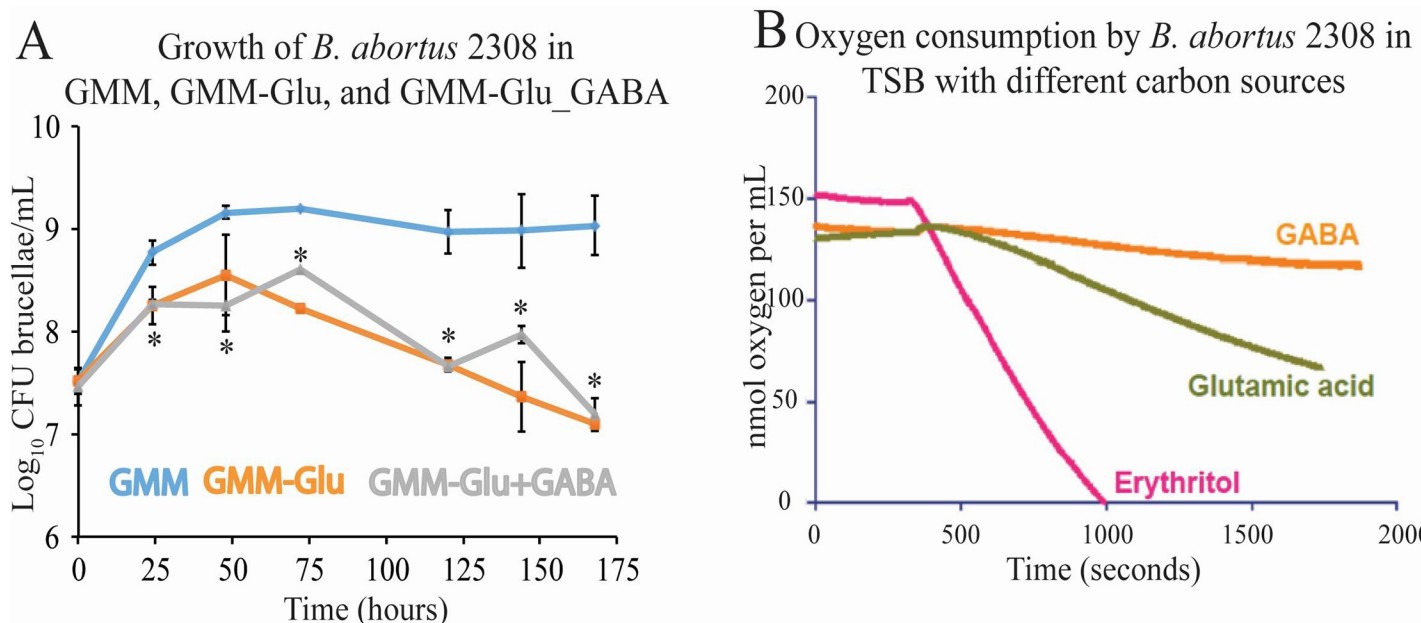

**Fig 7. GABA is not utilized as a metabolite by *B. abortus* 2308 *in vitro*.** A. Growth of *B. abortus* 2308 in minimal medium (GMM), minimal medium lacking glutamate (GMM-Glu), and minimal medium lacking glutamate with the addition of GABA (0.15%) (GMM-Glu+GABA). The asterisk denotes a statistically significant difference (* $P<0.05$; Student's t test) in uptake between *B. abortus* 2308 grown in GMM compared to *B. abortus* 2308 grown in either GMM-Glu or GMM-Glu+GABA. B. Oxygen consumption by *B. abortus* 2308 grown in TSB with the addition of either GABA, glutamate, or erythritol 300 seconds after inoculation measured via oxygraph machine.

A respirometry assay was utilized to assess GABA as a potential carbon source utilized by *B. abortus*. Oxygen concentrations of *B. abortus* cultures were measured via oxygraph machine to calculate respiration in response to different carbon sources. The carbon sources tested included GABA, glutamate, or erythritol. The metabolic role of glutamate is discussed above. Erythritol is a sugar alcohol found in the reproductive tracts of animals susceptible to brucellosis and has been shown to be a preferred carbon source for brucellae growth as well as an inducer of virulence related genes [33, 34]. Oxygen is consumed during aerobic respiration, thus if the bacterium is actively utilizing the supplied carbon source, then respiration will increase and oxygen concentrations of the culture medium will decrease. In the presence of erythritol, a preferred carbon source of *B. abortus*, respiration occurred at a high rate and oxygen levels decreased rapidly (Fig 7B). In the presence of glutamate, a suitable carbon and nitrogen source for *B. abortus*, but not preferred over erythritol, respiration occurred at a slower rate compared to erythritol, but oxygen consumption still occurred. In the presence of GABA, however, the change in oxygen concentration over time was negligible, indicating that GABA was not utilized as a carbon source by *B. abortus* (Fig 7B).

## Minimal transcriptional changes were observed in *B. abortus* 2308 exposed to exogenous GABA

RNAseq analysis was performed to assess the potential role of GABA as a signaling molecule. RNA was isolated from cultures of *B. abortus* 2308 grown aerobically in GMM in the presence or absence of 1 mM GABA and analyzed via RNA-seq analysis. Only one gene, the putative transposase BAB_RS29595, was downregulated 2.26 fold in the culture treated with 1 mM GABA.

## Discussion

In this study, the transport and biological function of GABA was analyzed in the intracellular pathogen *B. abortus*. The data presented revealed that $^3$H-GABA is transported under nutrient limiting conditions, transport was regulated by the sRNAs AbcR1 and AbcR2, and transport occurred via an ABC-transport system homologous to the *gts* system in *Rhizobium leguminosarum* and *A. tumefaciens*. Because of these results, we have annotated *bab2_0876-bab2_0879* as *gtsC*, *gtsB*, *gtsD*, *gtsA* for GABA transport system, respectively (Fig 1B).

The observation that the Gts system transports GABA preferentially over proteinogenic amino acids is intriguing, because this is not what is observed in *A. tumefaciens*. In *A. tumefaciens*, $^3$H-GABA uptake was competitively inhibited by short lateral chain amino acids, as well as proline [13]. Dissimilar from *A. tumefaciens*, $^3$H-GABA transport in *B. abortus* was uninhibited by the presence of other amino acids (Fig 3). This is a significant divergence between the two related organisms with regards to GABA transport and can be explained by a difference in the number of transporters between the two organisms. *A. tumefaciens* contains two GABA transport systems, Bra which is responsible for the transport of several amino acids and Gts which solely transports GABA [35]. As stated previously, *B. abortus* encodes two putative GABA transporters, *bab1_1794* which is homologous to the periplasmic binding protein of Bra and *bab2_0879* which is homologous to the periplasmic binding protein of Gts. Our results clearly show GABA transport in *B. abortus* is primarily mediated by *gts* (Fig 5). This result was surprising due to the low amino acid sequence identity of BAB2_0876-BAB2_0879 to the Gts GABA transporter in *R. leguminosarum* and *A. tumefaciens* (<40%) compared to the high amino acid sequence identity of BAB1_1794-BAB1_1799 compared to the Bra GABA transport system in *A. tumefaciens* (>70%) (Fig 1). In *B. abortus*, the Bra system (i.e., BAB1_1792–1799) does not appear to be a specific GABA transporter, and this reveals a deviation within

the *Rhizobiales* with regards to amino acid transport, which may explain why GABA transport was unaffected by the presence of other amino acids. Nonetheless, it is important to note that deletion of *bab2_0612* and *bab1_1792–1794* also lead to significant decreases in GABA transport (Fig 5), and therefore, while these systems may not be primary GABA transporters, it is clear that BAB2_0612 and BAB1_1792–1794 have the ability to support moderate GABA import by *B. abortus*.

We were unable to clearly define the biological role of GABA transport in *B. abortus*, despite metabolic and transcriptomic analyses. There are several potential reasons for this, including the potential for the identified transporter to also be able to transport other molecules. While the amino acid competition assay showed preference for GABA transport over proteinogenic amino acids, this does not discount the possibility that the Gts system is also able to transport other non-proteinogenic amino acids. The National Center for Biotechnology Information (NCBI) has *gtsA* annotated as a "spermidine/putrescine ABC transporter substrate-binding protein" in *B. abortus*. It has recently come to light that polyamines, such as spermidine and putrescine, are important for the persistence of *B. abortus* during chronic infection [36]. Therefore, further analyses of Gts in *B. abortus* may be necessary to characterize the potential for transport of both proteinogenic amino acids and polyamines.

Understanding the processing of imported GABA in other organisms can lead to important insights into the biological role of GABA; the *Brucella* genome may provide clues to this processing. The GABA shunt can be utilized to form succinate, a substrate utilized during the TCA cycle [2, 3]. This process occurs by converting GABA to succinic semialdehyde by the enzyme GABA-transaminase (GabT), followed by the conversion from succinic semialdehyde to succinate by succinic semialdehyde dehydrogenase (SSDH) [2]. Although the metabolic studies presented here do not reveal any metabolic utilization of GABA by *B. abortus*, the *Brucella* genome does contain genes encoding putative GabT (BAB2_0285) and SSDH (GabD, BAB1_1655). The function of these genes has not been characterized, but if functional, one or both could be important in the conversion of GABA to a utilizable carbon substrate. It is possible that mass spectrometry analyses of imported radiolabeled GABA could identify if and how GABA is processed by *B. abortus* and could lead to the formation of new hypotheses regarding the processing of this molecule by the brucellae. In the end, while we did not observe GABA utilization under the specific experimental conditions shown here, it is possible that *Brucella* catabolizes GABA under another condition or set of conditions that are encountered by the bacteria during infection.

This study and previous work from our lab has shown that deletion of *gtsA* (*bab2_0879*) did not change the ability of *B. abortus* to survival and replicate in peritoneally derived macrophage nor in a mouse model of infection via IP injection or oral route in BALB/c mice [32]. Interestingly, several studies identified *gtsA* as a potential virulence factor via different *in vitro* and *in vivo* screens. Firstly, the expression of *gtsA* has been shown to be increased in a quorum sensing mutant (Δ*babR*) and in the *B. melitensis* Rev. 1 vaccine strain when compared to the parental strain *B. melitensis* 16M [37, 38]. Genomic analysis revealed a nonsynonymous mutation in *gtsA* in the *B. melitensis* vaccine strain M5 when compared to *B. melitensis* 16M [39]. Delrue et al. published a list of attenuated *Brucella* mutants in large-scale *in vitro* and *in vivo* screens [40]. They reported that a mutation in BMEII0923 (a homolog of *gtsA* in *B. melitensis*) and BRA0326 (a homolog of *gtsA* in *B. suis*) was attenuated in a mouse model. Importantly, BAB2_0879, BMEII0923, and BRA0326 are 100% identical at the amino acid level. The type of mutation and mouse model utilized in this screen was not described, but this finding is intriguing and leads to further questions about whether *gtsA* has homologous functions in different *Brucella* species and whether the mouse model utilized in our study versus the Delrue study could lead to different observations in virulence [41]. Despite our results showing no

difference in infection between Δ*gtsA* compared to *B. abortus* 2308, these studies highlight the potential role for *gtsA* in *B. abortus* virulence and further analysis may be warranted. There are several possibilities for these differences, including differences in mouse strains (e.g., BALB/c vs. C57BL/6) and routes of infection (e.g., oral vs. intraperitoneal vs. aerosol) employed and moreover, while the *gts* system is dispensable for the infection of mice, it is possible that the Gts system is important for infection in natural host animals. Brucellosis also presents as a neurological disease in humans and marine mammals and recent evidence has shown that *B. abortus* can traverse the blood brain barrier [41–43]. Since GABA is an important and abundant neurotransmitter in the brain, it is possible that *gts* plays a significant and specific role in neurobrucellosis pathology.

Overall, this study characterizes the transport of the ubiquitous non-proteinogenic amino acid GABA by the intracellular bacterial pathogen *B. abortus*. While transport assays provided novel insights into the transport of this molecule by *B. abortus*, the data revealed limited metabolic, transcriptional, or virulence phenotypes in a deletion strain of this transporter. However, evidence found throughout scientific literature continues to add to the hypothesis that GABA and the Gts system are important for the pathogenesis of *Brucella*, warranting further studies to understand the biological role of GABA.

## Materials and methods

### Bacterial strains and growth conditions

*B. abortus* 2308 and derivative strains were routinely grown on Schaedler blood agar (SBA), which is composed of Schaedler agar (BD, Franklin Lakes, NJ) containing 5% defibrinated bovine blood (Quad Five, Ryegate, MT). Cultures were routinely grown in brucella broth (BD), tryptic soy broth (TSB), or in Gerhardt's Minimal Medium (GMM) [31]. For cloning, *Escherichia coli* strain DH5α was grown on tryptic soy agar (BD) or in Luria-Bertani (LB) broth. When appropriate, growth media were supplemented with kanamycin (45 μg/ml).

### Ethics statement

The experiments involving animals were carried out in strict accordance to the regulations set forth by Virginia Tech, as well as in accordance with all federal regulations. These experiments were approved by the Institutional Animal Care and Use Committee (IACUC) under Virginia Tech IACUC protocol 19–052. Additionally, these experiments were performed at Virginia Tech's VA-MD College of Veterinary Medicine, which is accredited by the Association for the Assessment and Accreditation of Laboratory Animal Care (AAALAC).

### Construction of *B. abortus* deletion strains

*B. abortus* strains containing isogenic, unmarked, nonpolar deletions of *bab2_0879* and *bab2_0612* were constructed for a previous study [32]. A single strain containing a deletion of the *bab1_1794* and *bab1_1792* locus in *B. abortus* 2308 was constructed using a nonpolar, unmarked gene excision strategy as described previously [44]. Briefly, an approximately 1-kb fragment of the upstream region of *bab1_1794* extending to the second codon of the coding region was amplified by PCR using primers *bab1_1794*-Up-For and *bab1_1794*-Up-Rev and genomic DNA from *B. abortus* 2308 as a template. Similarly, a fragment containing the last two codons of the coding region and extending to approximately 1 kb downstream of the *bab1_1792* open reading frame (ORF) was amplified with primers *bab1_1792*-Down-For and *bab1_1792*-Down-Rev. The sequences of all oligonucleotide primers used in this study can be found in Table 1, and the plasmids used in the study are listed in Table 2. The upstream

**Table 1. Oligonucleotide primers used in this study.**

| Primer name | Sequence (5'->3') |
|---|---|
| *bab1_1794*-Up-For | TAGGATCCTGTTCCCGCGTCTGAAGGAGC |
| *bab1_1794*-Up-Rev | GAAGGCGATGACTGCAGCAAGAG |
| *bab1_1792*-Down-For | TACTTCCAGAAGTAAATTGCC |
| *bab1_1792*-Down-Rev | GACTGCAGACGCTCAAAAAGATGGACCG |

*Underlined sequences depict a restriction endonuclease recognition site.

fragment was digested with *Bam*HI, the downstream fragment was digested with *Pst*I, and both fragments were treated with polynucleotide kinase in the presence of ATP. Both of the DNA fragments were included in a single ligation mix with *Bam*HI/*Pst*I-digested pNTPS138 (M.R.K. Alley, unpublished data) and T4 DNA ligase (Monserate Biotechnology Group, San Diego, CA). The resulting plasmid (pΔ*bab1_1794*Δ*bab1_1792*) was introduced into *B. abortus* 2308, and merodiploid transformants were obtained by selection on SBA plus kanamycin. A single kanamycin-resistant clone was grown for >6 h in brucella broth and then plated onto SBA containing 10% sucrose. Genomic DNA was isolated from sucrose resistant, kanamycin-sensitive colonies and screened by PCR for loss of the *bab1_1794-bab1_1792* locus.

## $^3$H-GABA uptake assays

A radiolabeled transport assay was utilized to assess the ability of *B. abortus* strains to import tritium labelled GABA ($^3$H-GABA) grown under several growth conditions. Gerhardt's Minimal Media (GMM) was inoculated with *Brucella* strains at a concentration of $10^9$ CFU brucellae/ml and incubated for 20 minutes at 37°C with shaking. The cultures were then inoculated with $^3$H-GABA at a final concentration of 100 nM and incubated for another 20 minutes at 37°C with shaking. The bacteria were collected via filtration through a filter (0.45 μm), washed three times with GMM, and the radioactivity of the filter was measured to quantify the amount of radiation imported by the brucellae collected on the filter by scintillation counter. If $^3$H-GABA is imported by *B. abortus*, then the filter will measure high radioactivity above background; however, if $^3$H-GABA is not imported by *B. abortus*, then the $^3$H-GABA will pass through the filter and the filter will not measure high radioactivity above background.

## Respirometry assay

Culture tubes of 5 mL of TSB were inoculated with *B. abortus* 2308 at a final concentration of $10^7$ CFU/mL and either 10 mM erythritol, glutamic acid, or GABA. The cultures were grown overnight at 37°C with shaking. The following day, the brucellae were pelleted, supernatant removed, and pellet resuspended in PBS at a final concentration of $10^2$ CFU/mL. Samples were then loaded into an oxygraph and oxygen concentrations were subsequently measured. After 300 seconds, erythritol, glutamic acid, or GABA were added to the corresponding culture tube at a final concentration of 100 mM and culture oxygen concentrations were measured for 2000 seconds.

**Table 2. Plasmids used in this study.**

| Plasmid name | Description | Reference |
|---|---|---|
| pNPTS138 | Cloning vector; contains *sacB*; Kan$^R$ | (M.R.K. Alley, unpublished) |
| pΔ*bab1_1794*Δ*bab1_1792* | In-frame deletion of *bab1_1794 and bab1_1792 locus* plus 1 kb of each flanking region in pNPTS138 | This study |

### RNA sampling of GABA treated cultures

Brucella broth was inoculated with *B. abortus* 2308 and incubated at 37˚C with shaking for ~24 hours until the cultured obtained an O.D. 600 nm of 0.15. Cells were then washed with PBS and cells were used to inoculate either GMM or GMM with the addition of 1 mM L-GABA at a concentration of $10^9$ CFU/mL. Cultures were incubated for 20 minutes at 37˚C with shaking. Following incubation, an equal volume of 1:1 ethanol:acetone was added to each culture and cultures were frozen at -80˚C until RNA isolation. This was performed in triplicate for each condition. RNA was isolated from each culture and DNase treated prior to submission for RNAseq analysis.

### Stranded RNA library construction for prokaryotic RNA-seq

1 μg of total RNA with RIN $\geq$ 8.0 was depleted of rRNA using Illumina's Ribo-Zero rRNA Removal Kit (Gram-Positive and Gram-Negative Bacteria) (P/N MRZB12424, Illumina, CA). The depleted RNA is fragmented and converted to first strand cDNA using reverse transcriptase and random primers using Illumina's TruSeq Stranded mRNA HT Sample Prep Kit (Illumina, RS-122-2103). This is followed by second strand synthesis using polymerase I and RNAse H, and dNTPs that contain dUTP instead of dTTP. The cDNA fragments then go through end repair, addition of a single 'A' base, and then ligation of adapters and indexed individually. The products are then purified and the second strand digested with N-Glycosylase, thus resulting in stranded template. The template molecules with the adapters are enriched by 10 cycles of PCR to create the final cDNA library. The library generated is validated using Agilent 2100 Bioanalyzer and quantitated using Quant-iT dsDNA HS Kit (Invitrogen) and qPCR. A total of 12 individually indexed cDNA libraries were pooled and sequenced on Illumina NextSeq.

### Illumina NextSeq sequencing

The libraries are clustered and sequenced using, NextSeq 500/550 High Output kit V2 (150 cycles) (P/N FC-404-2002) to 2 x 75 cycles to generate paired end reads. The Illumina NextSeq Control Software v2.1.0.32 with Real Time Analysis RTA v2.4.11.0 was used to provide the management and execution of the NextSeq 500 and to generate BCL files. The BCL files were converted to FASTQ files and demultiplexed using bcl2fastq Conversion Software v2.20.

### RNA-Seq data processing and analysis

The *B. abortus* 2308 gene and genome sequences, as well as corresponding annotations from NCBI (https://www.ncbi.nlm.nih.gov/) were used as the reference. Raw reads were quality-controlled and filtered with FastqMcf [45], resulting in an average of 1,821 Mbp (1,672 to 2,061 Mbp) nucleotides. The remaining reads were mapped to the gene reference using BWA (Li & Durbin, 2009) with default parameters. Differential expression of genes was calculated using the edgeR [46] package in R software (http://www.r-project.org/), with Benjamini–Hochberg adjusted P-values of 0.05 considered to be significant. The NCBI Sequence Read Archive (SRA) accession number for the RNA-seq data is PRJNA629010.

### Virulence of *Brucella* strains in cultured murine macrophages and experimentally infected mice

Experiments to test the virulence of *Brucella* strains in primary murine peritoneal macrophages were carried out as described previously [47]. Briefly, resident peritoneal macrophages were isolated from BALB/c mice and seeded in 96-well plates in Dulbecco's modified Eagle's

medium with 5% fetal bovine serum. The following day, *Brucella* strains were opsonized by incubating the strains with serum (1:1000 dilution) from previously infected mice (8 weeks post-infection) and the seeded macrophages were infected with brucellae at an MOI of 100:1. After 2 h of infection, extracellular bacteria were killed by treatment with gentamicin (50 μg/ml). For the 2-h time point, the macrophages were then lysed with 0.1% deoxycholate–PBS, and serial dilutions were plated on Schaedler blood agar (SBA). For the 24- and 48-h time points, the cells were washed with PBS following gentamicin treatment, and fresh cell culture medium containing gentamicin (20 μg/ml) was added to the monolayer. At the indicated time point, the macrophages were lysed, and serial dilutions were plated on SBA. Triplicate wells were used for each *Brucella* strain tested. Infection and colonization of mice by *Brucella* strains were measured by oral route of infection. BALB/c mice (6 mice per *Brucella* strain, 3 male and 3 female) were infected orally with $10^9$ CFU of each *Brucella* strain in sterile PBS. The animals were housed in microisolator cages in the ABSL3 laboratory, and the mice were subjected to 12-hour light– 12-hour dark cycles. Additionally, the mice were given access to pellet-style food and water ad libitum. The animals were monitored daily for signs of distress and pain in accordance with the guidelines of NIH's Animal Research Advisory Committee (https://oacu.oir.nih.gov/animal-research-advisory-committee-guidelines). The mice were sacrificed at 1, 2, and 4 weeks post-infection, and serial dilutions of spleen homogenates were plated on SBA to determine CFU counts of brucellae/spleen.

## Acknowledgments

We would like to thank the Teaching & Research Animal Care Support Service (TRACSS) at the VA-MD College of Veterinary Medicine for their rigorous and meticulous care of the animals used in this work.

## Author Contributions

**Conceptualization:** James A. Budnick, Clayton C. Caswell.

**Data curation:** James A. Budnick, Lauren M. Sheehan, Lin Kang, Pawel Michalak.

**Formal analysis:** James A. Budnick, Angela H. Benton, Joshua E. Pitzer, Lin Kang, Pawel Michalak, R. Martin Roop, II, Clayton C. Caswell.

**Funding acquisition:** Clayton C. Caswell.

**Investigation:** James A. Budnick, Lauren M. Sheehan.

**Methodology:** James A. Budnick, Lauren M. Sheehan, Angela H. Benton, Joshua E. Pitzer, Clayton C. Caswell.

**Project administration:** Clayton C. Caswell.

**Supervision:** Clayton C. Caswell.

**Validation:** James A. Budnick, Angela H. Benton, Joshua E. Pitzer, Clayton C. Caswell.

**Writing – original draft:** James A. Budnick.

**Writing – review & editing:** Clayton C. Caswell.

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
