## [Decision Letter · Decision Letter 0]

5 Jun 2020

PONE-D-20-12100

Characterizing the transport and utilization of the neurotransmitter GABA in the bacterial pathogen <brucella abortus="">

PLOS ONE

Dear Dr. Caswell,

Thank you for submitting your manuscript to PLOS ONE. After careful consideration, we feel that it has merit but does not fully meet PLOS ONE’s publication criteria as it currently stands. Therefore, we invite you to submit a revised version of the manuscript that addresses the points raised during the review process.</brucella>

Please consider the comments of the reviewers to improve the manuscript.

We look forward to receiving your revised manuscript.

Kind regards,

Axel Cloeckaert

Academic Editor

PLOS ONE

Journal Requirements:

2. In your Methods, please include full details of animal care and housing, including details of the monitoring of animals for adverse clinical signs.

Reviewers' comments:

Reviewer's Responses to Questions

**Comments to the Author**

1. Is the manuscript technically sound, and do the data support the conclusions?

Reviewer #1: Partly

Reviewer #2: Yes

Reviewer #3: Yes

2. Has the statistical analysis been performed appropriately and rigorously? 

Reviewer #1: I Don't Know

Reviewer #2: Yes

Reviewer #3: Yes

3. Have the authors made all data underlying the findings in their manuscript fully available?

Reviewer #1: Yes

Reviewer #2: No

Reviewer #3: Yes

4. Is the manuscript presented in an intelligible fashion and written in standard English?

Reviewer #1: Yes

Reviewer #2: Yes

Reviewer #3: Yes

5. Review Comments to the Author

Reviewer #1: The manuscript submitted by Budnick et al describes the efforts to characterized the putative role in GABA transport of two of the target transporter systems of sRNAs AbcR1/2, found previously by Dr. Caswell's group. By measuring accumulation of radioactive H3-GABA, the authors are able to provide some evidence of the role of one of them as a GABA transporter.

A drawing of the BAB1_1792-BAB1_1799 and BAB2_0876-BAB2_0879 transport systems, or a description of the role of its different components would be helpful to understand the experiments. Are the deleted proteins structural, crucial components, or just accessory ones? Similarly, a map of both locus would be helpful to know if the authors have deleted two or three genes, as it is not unusual in Brucella genomes to have non consecutive numbering. Figure 7 shows the locus bab2_0876-bab2_0879 along with the putative role of each protein, but this is too late in the manuscript and I think the reader would benefit from having this (or at least some of this) information up front, and included in the body of the manuscript.

Mention more clearly which strains were constructed for this work, as in page 11 is stated “Deletion strains of bab2_0612, bab1_1792-bab1_1794, and bab2_0879 were either constructed or utilized from previous studies”. As only deletion strains bab1_1792 and bab1_1794 are described in M&M, we have to assume that the other two strains were constructed in a previous study that should be referenced. This brings us to a shocking point, why have the authors mutated bab2_0612?? They do not mention anything about this strain, however it shows a 20% decrease in GABA transport.. Is it controlled by any of these sRNAs? Is it a control transporter? Without this information it’s impossible to understand its role, nor the role of the second transporter that also shows another 20% decrease in GABA transport. Regarding the deletion, and going back to the first comment, the reader does not get any information about the size or content of the region deleted, or the function of the affected proteins, until line 370, in the Discussion section, where the homology of one of the proteins is mentioned.

The 3H-GABA uptake assays is normalized to the observed level in GMM(-Glu). This should be explained in M&M, or in the Figure legend, rather than at the y axis of Figure 1. Same applies for Figure 2.

Regarding the physiological role of GABA transport, the authors observe that GABA is not used as a nitrogen or carbon source, and the transcriptional changes observed are anecdotal. Nor do they observe any change in virulence either in peritoneal macrophages or in the oral infection mouse model, reinforcing previous results from the group that these mutants do not show a different virulence levels in the intraperitoneal infection mouse model. But then they engaged in a not very convincing discussion about the possible role of these systems in virulence in B. abortus, citing some work done in B. melitensis and B. suis. It’s difficult with the data provided by the authors to gauge if this information is anything more than anecdotal. For example, how many nonsynonymous mutations are between the 16M and M5 B. melitensis strains? There are extensive areas of insertion and deletion between these two strains as to point out this particular change as important for the virulence. As for the attenuated phenotype reported by Delrue, the authors do not know the nature of the mutations described, and whether they could induce (for example) downstream effects. The authors should ponder if the level of evidence gathered so far merits further study of the putative role of gtsA in B. abortus, something that for this reviewer is in this moment doubtful.

The nomenclature bab_rs30470 is mentioned for the first time in page 16, without any explanation. The same way the authors mention that Agrobacterium fabrum was formerly known as Agrobacterium tumefaciens in page 4 (by the way, a reference to the change would be handy), the authors should use the old or the new nomenclature for the genes, or explain it at least the origin of this change.

Consistent nomenclature: The authors use B. abortus 2308, and Brucella melitensis biovar abortus 2308. This could be confusing for general readers outside the Brucella field, so please use consistent nomenclature.

Line 373. The results from Figure 4 merit a more extensive discussion. To say that GABA transport is mediated only by bab2_0879 is not correct. First, it should be mentioned the whole transporter, nor just the mutated gene that interrupts the system. And also the 20% (statistically significant, by the way) decrease in GABA transport observed when either bab2_0612 or bab1_1792-bab1_1794 are mutated should be considered and discussed.

Have the authors tested if B. abortus 2308 AbcR1 or AbcR2 single mutants show an inhibited GABA transport (just to see if in any occasion these sRNAs are not redundant…)

Minor points

Use B. abortus after the first time Brucella abortus is used

First three letters of restriction enzymes in italics

Keywords: amino

Reviewer #2: This manuscript describes the transport of GABA in the class III pathogen Brucella abortus. The transporter is identified and shown to be specific to GABA compared to 20 aminoacids. First investigations on the possible function of this transport system and the impact of GABA transport on the transcriptiome are reported here. This is an interesting, rigorous and well controlled study.

I only have (very) minor comments.

- line 76 : it seems that reference should be [16] and not [15]

- line 108 : "by adaptation" to what? intracellular conditions? [25] maybe specify

- line 126 : add one or more references after "GAD system"

- line 130 : why does the absence of a functional GAD excludes the possibility of GABA utilization? It seems that GabT and GabD would be sufficient to utilize GABA, if they are active of course.

- lines 187-191 : the authors comment on Cys and Lys but not the other aminoacids (Asn, Met,Trp and Val) that are also significantly different from the control. I agree that differences are very small and probably negligible

- line 235 : add a reference after "regulated by AbcR1 and AbcR2"

- line 243 : since a function is found for BAB2_0879, I would suggest to add a sentence to rename BAB2_0879 as gtsA at this place of the manuscript

- line 268-270 : the gtsA mutation does not impact the ability to survive and replicate in macrophages, but if glutamate is present in the culture medium of the macrophages, it is normal that the GABA transport system is not active (according to data in Fig. 1). If it is correct, this should be discussed

- lines 337-343 : if it is allowed by the journal, I would provide the mRNA quantification per gene as supplementary data

- line 359 : in the legend of Figure 7, I would clearly indicate which operon (bra or gts) is used for calculating the percentages of identity (from the text, it seems to be gts)

- since it is reported that gad gene may be inactivated by point or frame-shift mutations, it could be interesting to analyse the conservation of the gts, gabT and gabD genes in the different Brucella species.

- line 427 : for the information to these authors, the Delrue/Letesson lab and A. Dricot in particular were using exclusively the BALB/c mice, intraperiotoneal infections and CFU countings in the spleen (at 1, 4 or 8 weeks post-infection). Since it is not published (A. Dricot never finished her PhD thesis), I would leave the text as it is.

-line 434 : mouse strain is important but the route of infection is also crucial and should be mentionned here

- line 468 : if I understand correctly a single deletion strain is generated, in which ORFs bab1_1794 to bab1_1792 are deleted. Thus the sentence should be clarified. Also, a reference should be given for the deletion strains of bab2_0612 and bab2_0879 (and indicate if they are also unmarked and nonpolar deletion strains).

- line 501 : how was radioactivity of the filter measured?

- line 521 : is it L-GABA? I did not see an asymmetrical carbon the GABA structure

- line 548 : remove the sentence starting with "The FASTQ files"?

- line 567 : how were brucellae opsonized?

- Discussion : is there any comment to make, related to neurobrucellosis?

Reviewer #3: The paper by Budnick et. al. describes the identification of a GABA transporter in the zoonotic pathogen Brucella abortus. The authors show that B. abortus is able to transport GABA, that this transport is not inhibited by the presence of other amino acids and that GABA is not used neither as a nitrogen nor as a carbon source. The authors additionally demonstrate that the small RNAs AbcR1 and AbcR2 negatively regulate the transport of GABA and identify the transporter (Bab2_0879) as the responsible for the transport. A null mutation in this gene did not affect the intracellular replication in macrophages or the virulence in mice indicating that the transport is not necessary for the pathogenesis of the bacterium.

The paper is clear and the data supports the conclusions that are, mainly, negative: transport of GABA is not inhibited by other amino acids, it is not utilized as a carbon or nitrogen source, it is not required for virulence and does not affect the transcriptome of the bacterium. Despite this, the data presented clearly demonstrate that GABA is transported and that the transported is Bab2_0879.

The only criticisms I have is that in the discussion (lines 438 to 449) the authors argue that it could be that the transport of GABA could work as a way to mask the induction of GABA in the macrophages altering the maturation of the phagosome. If this would be the case a difference in macrophage survival should have been observed in the Bab2_0879 mutant. I recommend this speculation should be deleted.

6. PLOS authors have the option to publish the peer review history of their article (what does this mean?). If published, this will include your full peer review and any attached files.

Reviewer #1: No

Reviewer #2: No

Reviewer #3: No

---

## [Author Response · Author response to Decision Letter 0]

16 Jul 2020

Reviewer #1: The manuscript submitted by Budnick et al describes the efforts to characterized the putative role in GABA transport of two of the target transporter systems of sRNAs AbcR1/2, found previously by Dr. Caswell's group. By measuring accumulation of radioactive H3-GABA, the authors are able to provide some evidence of the role of one of them as a GABA transporter.

A drawing of the BAB1_1792-BAB1_1799 and BAB2_0876-BAB2_0879 transport systems, or a description of the role of its different components would be helpful to understand the experiments. Are the deleted proteins structural, crucial components, or just accessory ones? Similarly, a map of both locus would be helpful to know if the authors have deleted two or three genes, as it is not unusual in Brucella genomes to have non consecutive numbering. Figure 7 shows the locus bab2_0876-bab2_0879 along with the putative role of each protein, but this is too late in the manuscript and I think the reader would benefit from having this (or at least some of this) information up front, and included in the body of the manuscript.

This point is well taken, and we appreciate the suggestion. We agree completely with this comment, and as such, we have developed a new Fig 1 for the revised manuscript that depicts both the bab1_1792-1799 and bab2_0876-0879 loci. 

Mention more clearly which strains were constructed for this work, as in page 11 is stated “Deletion strains of bab2_0612, bab1_1792-bab1_1794, and bab2_0879 were either constructed or utilized from previous studies”. As only deletion strains bab1_1792 and bab1_1794 are described in M&M, we have to assume that the other two strains were constructed in a previous study that should be referenced. This brings us to a shocking point, why have the authors mutated bab2_0612?? They do not mention anything about this strain, however it shows a 20% decrease in GABA transport. Is it controlled by any of these sRNAs? Is it a control transporter? Without this information it’s impossible to understand its role, nor the role of the second transporter that also shows another 20% decrease in GABA transport. Regarding the deletion, and going back to the first comment, the reader does not get any information about the size or content of the region deleted, or the function of the affected proteins, until line 370, in the Discussion section, where the homology of one of the proteins is mentioned.

This too is a great point, and we apologize for the lack of information regarding these strains. The specific genes of interest were analyzed due to their significant dysregulation in the abcR1/abcR2 deletion stain [Ref 27]. Due to the significant increase in GABA transport observed in the abcR1/abcR2 deletion strain (Fig 4), we hypothesized that one of the ABC-type transport systems regulated by AbcR1/AbcR2 was functioning as a GABA transporter, and to test this hypothesis, we utilized deletion strains of the AbcR-regulation ABC transporter periplasmic-binding proteins BAB2_0612, BAB2_0879, and BAB1_1792-1794. The deletion strains of bab2_0612 and bab2_0879 were constructed previously (Ref. 32), and we have now noted this in the revised version of the manuscript. Bab1_1792 encodes a putative periplasmic-binding protein; bab1_1793 encodes a small hypothetical protein of 43 amino acids; bab1_1794 encodes a putative periplasmic-binding protein. Therefore, we decided to delete both genes encoding periplasmic-binding proteins (i.e., bab1_1792 and bab1_1794) in addition to the gene encoding the small hypothetical protein (i.e., bab1_1793) in order to determine if either of these proteins was involved in GABA transport by the ABC-type transporter that is encoded nearby on the chromosome. In the end, there was a slight reduction in GABA transport in both the bab2_0612 and bab1_1792-1794 deletion strains, but the most significant decrease in GABA uptake was observed for the bab2_0879 deletion strain. We have included more informaiton about the strains in the Results section of the revised manuscript (Lines 257-262). 

The 3H-GABA uptake assays is normalized to the observed level in GMM(-Glu). This should be explained in M&M, or in the Figure legend, rather than at the y axis of Figure 1. Same applies for Figure 2.

Thank you for this suggestion, and we have included this information in the figure legend of the figures of the revised manuscript. Please note these figures are now Fig 2 and Fig 3 in the revised manuscript.

Regarding the physiological role of GABA transport, the authors observe that GABA is not used as a nitrogen or carbon source, and the transcriptional changes observed are anecdotal. Nor do they observe any change in virulence either in peritoneal macrophages or in the oral infection mouse model, reinforcing previous results from the group that these mutants do not show a different virulence levels in the intraperitoneal infection mouse model. But then they engaged in a not very convincing discussion about the possible role of these systems in virulence in B. abortus, citing some work done in B. melitensis and B. suis. It’s difficult with the data provided by the authors to gauge if this information is anything more than anecdotal. For example, how many nonsynonymous mutations are between the 16M and M5 B. melitensis strains? There are extensive areas of insertion and deletion between these two strains as to point out this particular change as important for the virulence. As for the attenuated phenotype reported by Delrue, the authors do not know the nature of the mutations described, and whether they could induce (for example) downstream effects. The authors should ponder if the level of evidence gathered so far merits further study of the putative role of gtsA in B. abortus, something that for this reviewer is in this moment doubtful.

These points are very well taken, and we appreciate the careful review of our data and the current literature. We agree that the virulence data from our work with B. abortus and others’ work with other Brucella strains are variable, and in fact, we are only trying to convey to readers that these differences have been observed. Many researchers in the Brucella field utilize mice as a model, but we all agree that the mouse is not an appropriate model for the natural hosts (e.g., cattle, goats, ect.). As such, we have attempted to underscore the differences that our group and others have observed with the mouse model as it relates to the Gts system. The Reviewer is correct that we do not know all of the details about the experiments reported by Delrue and colleagues, and we have included additional discussion to this point. It is possible that differences in mouse strains and/or route of infection have played a role in the observed differences, and we have noted this in the revised version of the manuscript on lines 454-461.

Also, to the point about differences in GtsA between B. abortus, B. melitensis, and B. suis, we have noted in the revised version of the manuscript that GtsA is 100% identical at the amino acid levels between these strains (lines 448-449).

Finally, we also agree that we used strong language expressing that the Gts systems should be studied further based on our data and the data of others. As such, we now suggest that the Gts system “may” be something that warrants further investigation (line 456). We believe this is an accurate reflection of the situation, particularly since the Gts system has not been evaluated in natural hosts.

The nomenclature bab_rs30470 is mentioned for the first time in page 16, without any explanation. The same way the authors mention that Agrobacterium fabrum was formerly known as Agrobacterium tumefaciens in page 4 (by the way, a reference to the change would be handy), the authors should use the old or the new nomenclature for the genes, or explain it at least the origin of this change. 

Consistent nomenclature: The authors use B. abortus 2308, and Brucella melitensis biovar abortus 2308. This could be confusing for general readers outside the Brucella field, so please use consistent nomenclature.

These are great points, and we agree with the Reviewer 100% about the ambiguity of both the locus tags in Brucella and the Agrobacterium nomenclature. We will first discuss the Agrobacterium issue. We have learned from our research that the suggested change from A. tumefaciens to A. fabrum is not well supported in the literature, and as such, we have removed all references to A. fabrum from the revised version of the manuscript. We now refer only to A. tumefaciens.

Regarding the Brucella abortus 2308, it is admittedly convoluted; however, we are trying our best to inform the readers of exactly what strain we are referring to in our studies. Several years ago, it was suggested that many Brucella strains are in fact the same species, Brucella melitensis, and that what was previously used as other species names (e.g., abortus) were actually biovars. During this time, the genome sequence and annotation for the reference strain was published, and the name given to it was “Brucella melitensis biovar Abortus 2308.” This species/biovar decision was later reversed, but the “Brucella melitensis biovar Abortus 2308” designation has remained in NCBI to this day. To make the situation more confusing, there is another genome sequence for a strain designated “Brucella abortus S2308,” which is slightly different from the “Brucella melitensis biovar Abortus 2308” that many labs use. Therefore, in order to direct readers to the appropriate genome, we have in certain places (e.g., regarding the RNA-seq data) referred to “Brucella melitensis biovar Abortus 2308,” but overall, those in the field know this strain as “Brucella abortus 2308.”

There are similar issues with the locus tags of “Brucella melitensis biovar Abortus 2308,” as the Reviewer aptly pointed out. The original locus tags were “BAB1_####” and “BAB2_####,” where 1 and 2 refer to chromosome 1 or 2, and the # was the order of the gene on the chromosome. About 6-7 years ago, the genomes were re-annotated and genes given the “BAB_RS#####” designation. Unfortunately, there is no link between BAB and BAB_RS locus tags. Therefore, we, and many others in the field, have adopted a practice of including both the BAB and BAB_RS designations when first discussing a particular gene in the literature, as this allows researcher to use both current or more dated databases and papers to analyze the data. To the Reviewer’s specific point, we have now made sure that we introduce the BAB and BAB_RS nomenclature at the same time, and then only refer to the BAB designations subsequently. In terms of bab_rs30470, it is now introduced in the revised version of the manuscript in line 128.

Line 373. The results from Figure 4 merit a more extensive discussion. To say that GABA transport is mediated only by bab2_0879 is not correct. First, it should be mentioned the whole transporter, nor just the mutated gene that interrupts the system. And also the 20% (statistically significant, by the way) decrease in GABA transport observed when either bab2_0612 or bab1_1792-bab1_1794 are mutated should be considered and discussed.

Thank you for this really insightful point, and we agree with the Reviewer on this point. Firstly, it was incorrect for us to say that BAB2_0879 is the “only” transporter for GABA, as our data clearly show this is not the case. We have revised this language to say that the Gts system is the “primary” GABA transporter in B. abortus 2308 (line 392). Secondly, we fully agree that more discussion about the role of BAB2_0612 and BAB1_1792-1794 in GABA import is warranted, and we have added more discussion in the revised manuscript (lines 400-404).

Have the authors tested if B. abortus 2308 AbcR1 or AbcR2 single mutants show an inhibited GABA transport (just to see if in any occasion these sRNAs are not redundant…)

This is a great suggestion, but unfortunately, we have not been able to test GABA transport in the abcR1 and abcR2 single deletion strains. However, we have shown previously that gtsA (i.e., bab2_0879 is very effectively regulated by either AbcR1 or AbcR2, and bab1_1794 and bab2_0612 are likewise regulated by either AbcR1 or AbcR2 [Ref 32]. As such, it is likely that GABA transport will not be significantly affected by single deletion of abcR1 or abcR2.

Minor points

Use B. abortus after the first time Brucella abortus is used

First three letters of restriction enzymes in italics

Thank you for pointing these things out, and we have corrected these issues. We now use “B. abortus” after the first introduction of “Brucella abortus.” Additionally, we have formatted the enzyme designations accordingly.

Keywords: amino

We appreciate the discovery of this mistake, and we have corrected it.

Reviewer #2: This manuscript describes the transport of GABA in the class III pathogen Brucella abortus. The transporter is identified and shown to be specific to GABA compared to 20 aminoacids. First investigations on the possible function of this transport system and the impact of GABA transport on the transcriptiome are reported here. This is an interesting, rigorous and well controlled study.

I only have (very) minor comments.

- line 76 : it seems that reference should be [16] and not [15]

Thank you very much for catching our mistake. We have corrected this.

- line 108 : "by adaptation" to what? intracellular conditions? [25] maybe specify

This point is well taken, and we have revised the statement accordingly. “Adaptation” was meant to refer to the harsh intracellular environment that the brucellae encounter following their uptake by host cells, and we have added this to our description of adaptation in the revised manuscript (lines 108-109).

- line 126 : add one or more references after "GAD system"

We appreciate this suggestion, and we have now added a reference for the GAD system.

- line 130 : why does the absence of a functional GAD excludes the possibility of GABA utilization? It seems that GabT and GabD would be sufficient to utilize GABA, if they are active of course.

This is a great point, and we agree that the absence of GAD does not exclude the possible utilization of GABA. Therefore, in the Introduction, we now state that the “potential role of GABA utilization…is unknown” (lines 149-150). We further elaborate on the possibility that GABA is utilized using GabT and GabD in the Discussion of the revised manuscript (lines 418-435).

- lines 187-191 : the authors comment on Cys and Lys but not the other aminoacids (Asn, Met,Trp and Val) that are also significantly different from the control. I agree that differences are very small and probably negligible

Thank you for pointing out this omission. The Reviewer is entirely correct, and we have added discussion of these other amino acids in the Results section (lines 206-212).

- line 235 : add a reference after "regulated by AbcR1 and AbcR2"

This is a great suggestion, and we have now added the appropriate citations.

- line 243 : since a function is found for BAB2_0879, I would suggest to add a sentence to rename BAB2_0879 as gtsA at this place of the manuscript

Thank you for this great suggestion, and have defined BAB2_0879 as GtsA at this point in the revised manuscript (lines 269-271).

- line 268-270 : the gtsA mutation does not impact the ability to survive and replicate in macrophages, but if glutamate is present in the culture medium of the macrophages, it is normal that the GABA transport system is not active (according to data in Fig. 1). If it is correct, this should be discussed

This too is a really great point, and we have further investigated the composition of our medium. For the macrophage phage experiments, we utilize Dulbecco’s modified Eagle’s medium, and while DMEM contains a variety of amino acids, it does not contain glutamate. Nonetheless, it is possible that expression and/or activation of the Gts system is affected during these assays, but the presence of glutamate is likely not the reason.

- lines 337-343 : if it is allowed by the journal, I would provide the mRNA quantification per gene as supplementary data

Thank you for this suggestion. Given that only one gene was differentially expressed in response to GABA, we did not feel it was justified to include supplemental data for all of the reads from the RNA-seq experiment. However, we have deposited all of the RNA-seq data, and it is freely available to the community. The accession number is PRJNA629010 (line 579). 

- line 359 : in the legend of Figure 7, I would clearly indicate which operon (bra or gts) is used for calculating the percentages of identity (from the text, it seems to be gts)

This is a great point, and we have developed a new figure to show the identity of both systems to the corresponding systems in B. abortus. Please note that the new figure is now Fig 1, as it was suggested by another Reviewer that this information would be helpful if presented earlier in the manuscript.

- since it is reported that gad gene may be inactivated by point or frame-shift mutations, it could be interesting to analyse the conservation of the gts, gabT and gabD genes in the different Brucella species.

Thank you for this suggestion, and we agree that this analysis is quite interesting. We have accessed the GtsA orthologs in B. abortus 2308 (i.e., BAB2_0879), B. melitensis 16M (i.e., BMEII0923), and B. suis 1330 (i.e., BRA0326), and interestingly, these proteins are 100% identical at the amino acid level. We have included this information in the Discussion section of the revised manuscript (lines 448-449).

- line 427 : for the information to these authors, the Delrue/Letesson lab and A. Dricot in particular were using exclusively the BALB/c mice, intraperiotoneal infections and CFU countings in the spleen (at 1, 4 or 8 weeks post-infection). Since it is not published (A. Dricot never finished her PhD thesis), I would leave the text as it is.

Thank you very much for sharing this information with us, and as the Reviewer suggested, given that the information is unpublished, we did not alter the text from the original submission.

-line 434 : mouse strain is important but the route of infection is also crucial and should be mentionned here

This is a fantastic point, and we have included a discussion of the infection route in the revised manuscript (lines 458-459).

- line 468 : if I understand correctly a single deletion strain is generated, in which ORFs bab1_1794 to bab1_1792 are deleted. Thus the sentence should be clarified. Also, a reference should be given for the deletion strains of bab2_0612 and bab2_0879 (and indicate if they are also unmarked and nonpolar deletion strains).

We appreciate these suggestions, and we have further clarified these points. It was noted by another Reviewer that this information should be included in the Results section as well. Therefore, we have clarified the strain construction along with citations in the revised manuscript (lines 257-262 and in lines 484-488).

- line 501 : how was radioactivity of the filter measured?

We apologize for this omission, and we are grateful to the Reviewer for catching our mistake. The radioactivity was measured using a scintillation counter, and this is now described in line 522.

- line 521 : is it L-GABA? I did not see an asymmetrical carbon the GABA structure

This was an oversight on our part, and we thank the Reviewer for pointing out this issue. All of the amino acids were used in the study for competition assays were L-form amino acids. However, GABA does not exist in L- and D-forms, and as such, we have corrected any mention of “L-GABA” in the revised manuscript.

- line 548 : remove the sentence starting with "The FASTQ files"?

Thank you for this suggestion. We have removed the sentence.

- line 567 : how were brucellae opsonized?

This is a great question, and the bacteria were opsonized with serum from mice previously infected with B. abortus 2308. We have included this information in the Materials and Methods section of the revised manuscript (lines 586-588).

- Discussion : is there any comment to make, related to neurobrucellosis?

This is a wonderful question, and we agree with the Reviewer that this is an interesting subject. As such, we have included a discussion of neurobrucellosis in the Discussion section of the revised manuscript (lines 461-465).

Reviewer #3: The paper by Budnick et. al. describes the identification of a GABA transporter in the zoonotic pathogen Brucella abortus. The authors show that B. abortus is able to transport GABA, that this transport is not inhibited by the presence of other amino acids and that GABA is not used neither as a nitrogen nor as a carbon source. The authors additionally demonstrate that the small RNAs AbcR1 and AbcR2 negatively regulate the transport of GABA and identify the transporter (Bab2_0879) as the responsible for the transport. A null mutation in this gene did not affect the intracellular replication in macrophages or the virulence in mice indicating that the transport is not necessary for the pathogenesis of the bacterium.

The paper is clear and the data supports the conclusions that are, mainly, negative: transport of GABA is not inhibited by other amino acids, it is not utilized as a carbon or nitrogen source, it is not required for virulence and does not affect the transcriptome of the bacterium. Despite this, the data presented clearly demonstrate that GABA is transported and that the transported is Bab2_0879.

The only criticisms I have is that in the discussion (lines 438 to 449) the authors argue that it could be that the transport of GABA could work as a way to mask the induction of GABA in the macrophages altering the maturation of the phagosome. If this would be the case a difference in macrophage survival should have been observed in the Bab2_0879 mutant. I recommend this speculation should be deleted.

We are very grateful to the Reviewer for their comments and suggestions. We agree that it may be premature to speculate on the potential masking of GABA in macrophages, and as such, we have removed this paragraph from the Discussion section of the revised manuscript.

---

## [Editor Report · Decision Letter 1]

5 Aug 2020

Characterizing the transport and utilization of the neurotransmitter GABA in the bacterial pathogen <brucella abortus="">

PONE-D-20-12100R1</brucella>

Dear Dr. Caswell,

We’re pleased to inform you that your manuscript has been judged scientifically suitable for publication and will be formally accepted for publication once it meets all outstanding technical requirements.

Kind regards,

Axel Cloeckaert

Academic Editor

PLOS ONE
---

## [Editor Report · Acceptance letter]

5 Aug 2020

PONE-D-20-12100R1 

Characterizing the transport and utilization of the neurotransmitter GABA in the bacterial pathogen <Brucella abortus> 

Dear Dr. Caswell:

I'm pleased to inform you that your manuscript has been deemed suitable for publication in PLOS ONE. Congratulations! Your manuscript is now with our production department. 

Kind regards, 

on behalf of

Dr. Axel Cloeckaert 

Academic Editor

PLOS ONE